# Fractal Generative Models

**Tianhong Li**                                             *tianhong@mit.edu*
*MIT*

**Qinyi Sun**                                               *wendysun@mit.edu*
*MIT*

**Lijie Fan**                                               *lijiefan@google.com*
*Google DeepMind*

**Kaiming He**                                              *kaiming@mit.edu*
*MIT*

**Reviewed on OpenReview:** *https://openreview.net/forum?id=Qk9kn6lOlW*

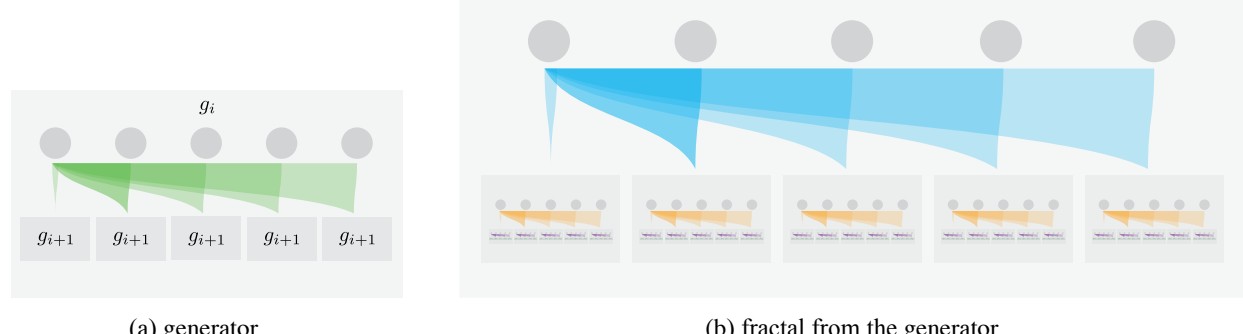

(a) generator                                    (b) fractal from the generator

Figure 1: **Fractal Generative Model**. Four levels shown in this figure, better viewed zoomed in. In this instantiation, we use autoregressive model for the fractal generator. By recursively calling autoregressive models in autoregressive models, we build a fractal-like framework with self-similarity across different levels.

## Abstract

Modularization is a cornerstone of computer science, abstracting complex functions into atomic building blocks. In this paper, we introduce a new level of modularization by abstracting generative models themselves into atomic modules. Our method constructs generative models by recursively invoking atomic generative modules, resulting in architectures with fractal-like, self-similar properties. We call this new class of models ***fractal generative models***. As a running example, we instantiate our fractal framework using autoregressive models as the atomic modules and examine it on the challenging task of pixel-by-pixel image generation. Our experiments show strong performance in both likelihood estimation and generation quality. We hope this work could serve as a starting point for future research into fractal generative models, establishing a new paradigm in generative modeling.

## 1 Introduction

At the core of computer science lies the concept of modularization. For example, deep neural networks are built from atomic "layers" that serve as modular units (Szegedy et al., 2015). Similarly, modern generative models—such as diffusion models (Song et al., 2020) and autoregressive models (Radford et al., 2018)—are built from atomic "generative steps", each implemented by a deep neural network. By abstracting complex functions into these atomic building blocks, modularization allows us to create more intricate systems by composing these modules.

Building on this concept, we propose abstracting a generative model itself as a module to develop more advanced generative models. Specifically, we introduce a generative model constructed by recursively invoking generative models of the same kind within itself. This recursive strategy results in a generative framework that exhibits complex architectures with self-similarity across different levels of modules, as shown in Figure 1.

Our concept draws inspiration from the mathematical notion of **_fractals_** (Mandelbrot, 1983). While classical fractals exhibit exact self-similarity at each recursion level, our framework adopts a looser, conceptual interpretation: each recursion involves generative modules of the same type but not necessarily sharing identical parameters. This relaxed definition provides greater flexibility, allowing each fractal level to adapt to the specific data distributions at different scales, a key advantage in modeling complex, real-world data. Reflecting this conceptual analogy, we term our framework "**_fractal generative models_**".

This conceptual interpretation also aligns well with fractal-like patterns widely observed in nature. Biological neural networks, for example, frequently exhibit fractal-like, scale-invariant connectivity (Bassett et al., 2006; Sporns, 2006; Bullmore & Sporns, 2009; Ansell & Kovács, 2024), suggesting that the brain's development largely adopts the concept of modularization, recursively building larger neural networks from smaller ones. Likewise, fractal patterns are prevalent in natural data, ranging from macroscopic structures such as clouds, tree branches, and snowflakes, to microscopic ones including crystals (Cannon et al., 2000), chromatin (Mirny, 2011), and proteins (Enright & Leitner, 2005). More generally, natural images can also be viewed through a fractal lens—an image is composed of sub-images that are themselves images (although they may follow different distributions). Thus, it is natural to consider generative models whose modules recursively invoke similar generative functionalities.

In this work, we instantiate our fractal generative framework using autoregressive modules as the _generator_[1] which defines the recursive generation rule (Figure 1). In our instantiation, each autoregressive model is composed of modules that are themselves autoregressive models. Concretely, each parent autoregressive block, modeling a high-dimensional data distribution, recursively spawns multiple child autoregressive blocks, each modeling lower-dimensional distributions, and each child block further spawns more autoregressive blocks. The resulting hierarchical structure thus exhibits fractal-like, self-similar patterns across multiple levels.

We examine this fractal instantiation on a challenging testbed: pixel-by-pixel image generation. Existing methods that directly model the pixel sequence do not achieve satisfactory results in both likelihood estimation and generation quality (Hawthorne et al., 2022; Yu et al., 2023), as images do not embody a clear sequential order. Despite its difficulty, pixel-by-pixel generation represents a broader class of important generative problems: modeling non-sequential data with intrinsic structures, which is particularly important for many data types beyond images, such as molecular structures, proteins, and biological neural networks.

Our proposed fractal framework demonstrates strong performance on this challenging yet important task. It can generate raw images pixel-by-pixel (Figure 3) while achieving accurate likelihood estimation and high generation quality. We hope our promising results will encourage further explorations of fractal generative models in their design and applications, ultimately establishing a new paradigm for generative modeling.

## 2    Related Work

**Fractals.** A _fractal_ is a geometric structure characterized by self-similarity across different scales, often constructed by a recursive generation rule called a generator (Mandelbrot, 1983). Fractals are widely observed in nature, with classic examples ranging from macroscopic structures such as clouds, tree branches, and snowflakes, to microscopic ones including crystals (Cannon et al., 2000), chromatin (Mirny, 2011), and proteins (Enright & Leitner, 2005).

Beyond these more easily recognizable fractals, many natural data also exhibit near-fractal characteristics. Although they do not possess strict self-similarity, they still embody similar multi-scale representations

---

[1]We use the term "generator" following Mandelbrot (1983). In this paper, "generator" specifically refers to a rule for recursively forming fractals.

or patterns, such as images (Freeman et al., 1991; Lowe, 1999) and biological neural networks (Bassett et al., 2006; Sporns, 2006; Bullmore & Sporns, 2009). Conceptually, our fractal generative model naturally accommodates all such non-sequential data with intrinsic structures and self-similarity across different scales; in this paper, we demonstrate its capabilities with an image-based instantiation.

**Hierarchical Representations.** Extracting hierarchical pyramid representations from visual data has long been an important topic in computer vision. Many early hand-engineered features, such as Steerable Filters, Laplacian Pyramid, and SIFT, employ scale-space analysis to construct feature pyramids (Burt & Adelson, 1987; Freeman et al., 1991; Lowe, 1999; 2004; Dalal & Triggs, 2005). In the context of neural networks, hierarchical designs remain important for capturing multi-scale information. For instance, SPPNet (He et al., 2015) and FPN (Lin et al., 2017) construct multi-scale feature hierarchies with pyramidal feature maps. Our fractal framework is also related to Swin Transformer (Liu et al., 2021), which builds hierarchical feature maps by attending to local windows at different scales. These hierarchical representations have proven effective in various image understanding tasks, including image classification, object detection, and semantic segmentation.

**Hierarchical Generative Models.** Hierarchical designs are also widely used in generative modeling. Many recent methods employ a two-stage paradigm, where a pre-trained tokenizer maps images into a compact latent space, followed by a generative model on those latent codes (van den Oord et al., 2017; Razavi et al., 2019; Esser et al., 2021; Ramesh et al., 2021). Another example, MegaByte (Yu et al., 2023), implements a two-scale model with a global and a local module for more efficient autoregressive modeling of long pixel sequences, though its performance remains limited. Some recent work also applies such hierarchical concepts in reasoning models and RNNs (Wang et al., 2025; Nzoyem et al., 2025).

Another line of research focuses on scale-space image generation. Cascaded generative models (Yu et al., 2020; Ramesh et al., 2022; Ho et al., 2022; Pernias et al., 2023) train multiple generative models, such as diffusion models or normalizing flow models, to progressively generate images from low resolution to high resolution. More recently, scale-space autoregressive methods (Tian et al., 2024; Tang et al., 2024; Han et al., 2024) generate tokens one scale at a time using an autoregressive transformer. However, generating images without a tokenizer is often prohibitively expensive for these autoregressive approaches because the large number of tokens or pixels per scale leads to a quadratic computational cost for the attention within each scale.

Our fractal generative model builds upon hierarchical generative principles but uniquely integrates recursive, self-similar structures inspired by fractal geometry. Unlike conventional hierarchical models that typically use multiple stages at fixed resolutions, our fractal framework recursively invokes generative modules of the same type across multiple image scales, analogous to the recursive self-similarity of natural fractal structures. This recursive design enables our approach to effectively capture intrinsically structured, non-sequential data, introducing a novel form of hierarchical generative modeling.

**Modularized Neural Architecture Design.** Modularization is a fundamental concept in computer science and deep learning, which atomizes previously complex functions into simple modular units. One of the earliest modular neural architectures was GoogleNet (Szegedy et al., 2015), which introduced the "Inception module" as a new level of organization. Later research expanded on this principle, designing widely used units such as the residual block (He et al., 2016) and the Transformer block (Vaswani, 2017). Recently, in the field of generative modeling, MAR (Li et al., 2024) modularize diffusion models as atomic building blocks to model the distribution of each continuous token, enabling the autoregressive modeling of continuous data. By providing higher levels of abstraction, modularization enables us to build more intricate and advanced neural architectures using existing methods as building blocks.

A pioneering approach that applies a modular unit recursively and integrates fractal concepts in neural architecture design is FractalNet (Larsson et al., 2016), which constructs very deep neural networks by recursively calling a simple expansion rule. While FractalNet shares our core idea of recursively invoking a modular unit to form fractal structures, it differs from our method in two key aspects. First, FractalNet uses a small block of convolutional layers as its modular unit, while we use an entire generative model, representing different levels of modularization. Second, FractalNet was mainly designed for classification tasks and thus outputs only low-dimensional logits. In contrast, our approach leverages the exponential scaling behavior of

fractal patterns to generate a large set of outputs (e.g., millions of image pixels), demonstrating the potential of fractal-inspired designs for more complex tasks beyond classification.

# 3 Fractal Generative Models

In this section, we introduce the fractal generative framework and illustrate how it can efficiently model high-dimensional data distributions by recursively composing atomic generative modules. We first outline the general framework and then present an instantiation using autoregressive models as the atomic modules.

## 3.1 Fractal Generative Framework

A fractal generative model recursively builds more powerful generative models from simpler atomic generative modules. Formally, each fractal level consists of a generator $g_i$ that transforms an output $x_i$ from the previous level into a set of new outputs $\{x_{i+1}\}$: $\{x_{i+1}\} = g_i(x_i)$. By recursively invoking similar generative modules at each level (as illustrated in Figure 1), this framework achieves exponential output growth with only a linear increase in recursive depth. This property enables efficient modeling of complex, high-dimensional data.

Unlike classical fractals with strict self-similarity and parameter sharing, our framework employs a flexible, conceptual fractal structure. Each fractal generator level uses independently parameterized modules, allowing the model to adapt to scale-specific data characteristics while keeping structural self-similarity across recursive levels.

## 3.2 Instantiation with Autoregressive Modules

We demonstrate the fractal generative framework by instantiating it with autoregressive models. Our goal is to model the joint distribution of a large set of random variables $x_1, \cdots, x_N$, but directly modeling it with a single autoregressive model is computationally prohibitive. To address this, we adopt a *divide-and-conquer* strategy. The key modularization is to abstract an autoregressive model as a modular unit that models a conditional probability distribution $p(x|c)$. With this modularization, we can construct a more powerful autoregressive model by building it on top of multiple next-level autoregressive models.

Specifically, assume each autoregressive module handles sequences of length $k$, and the total number of variables is $N = k^n$, where $n = \log_k(N)$ represents the number of recursive levels in our fractal framework. The joint distribution factorizes recursively as:

$$p(x_1, \cdots, x_{k^n}) = \prod_{i=1}^{k} p(x_{(i-1) \cdot k^{n-1}+1}, \cdots, x_{i \cdot k^{n-1}} | x_1, \cdots, x_{(i-1) \cdot k^{n-1}}),$$

where each conditional distribution $p(\cdots | \cdots)$ with $k^{n-1}$ variables is modeled by an autoregressive module at the next fractal level. By recursively calling such a divide-and-conquer process, our fractal framework can efficiently handle the joint distribution of $k^n$ variables using $n$ levels of autoregressive models, each operating on a manageable sequence length $k$.

This recursive process represents a standard divide-and-conquer strategy. By recursively decomposing the joint distribution, our fractal autoregressive architecture not only significantly reduces computational costs compared to a single large autoregressive model but also captures the intrinsic hierarchical structure within the data.

Conceptually, as long as the data exhibits a structure that can be organized in a divide-and-conquer manner, it can be naturally modeled within our fractal framework. To provide a more concrete example, in the next section, we implement this approach to tackle the challenging task of pixel-by-pixel image generation.

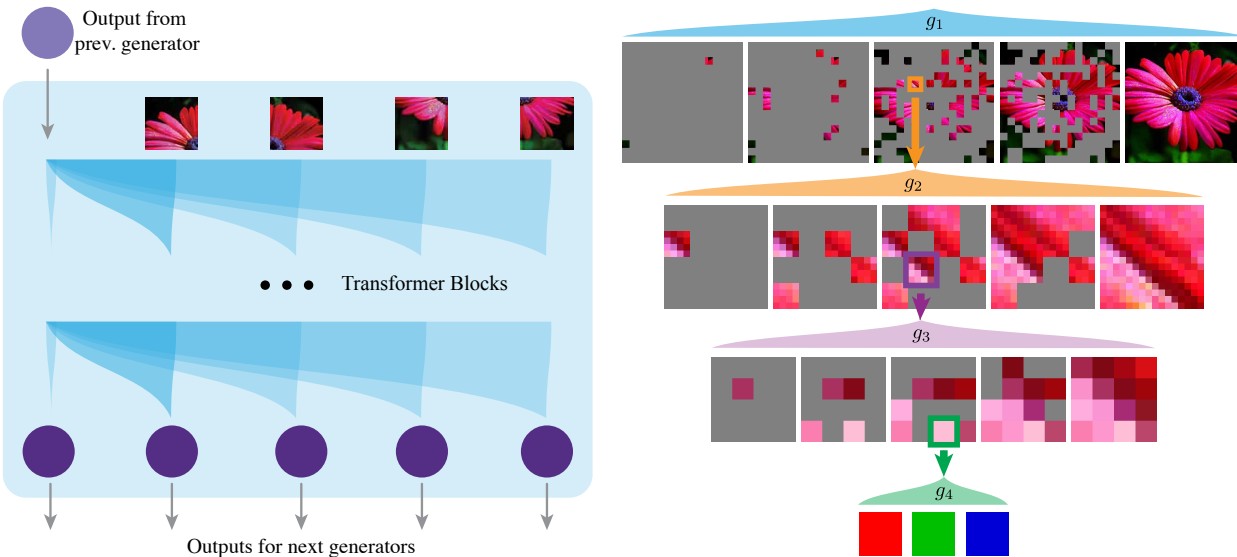

Figure 2: **Instantiation of our fractal method on pixel-by-pixel image generation.** In each fractal level, an autoregressive model receives the output from the previous generator, concatenates it with the corresponding image patches, and employs multiple transformer blocks to produce a set of outputs for the next generators.

Figure 3: **Our fractal framework can generate high-quality images pixel-by-pixel.** We show the generation process of a 256×256 image by recursively calling autoregressive models in autoregressive models. We also provide additional example videos in the supplementary material to illustrate the generation process.

Table 1: **Model configurations, parameters, and computational costs** at different autoregressive levels for our large-size model. The computational costs are measured by GFLOPs per forward pass during training with batch size 1. Notably, thanks to the fractal design, the total computational cost of modeling a 256×256 image is only twice that of modeling a 64×64 image.

| image size | | seq. len. | #layers | hidden dim | #params (M) | #GFLOPs |
|---|---|---|---|---|---|---|
| | $g_1$ | 16×16 | 32 | 1024 | 403 | 215 |
| 64×64×3 | $g_2$ | 4×4 | 8 | 512 | 25 | 208 |
| | $g_3$ | 3 | 3 | 128 | 0.6 | 15 |
| | $g_1$ | 16×16 | 32 | 1024 | 403 | 215 |
| 256×256×3 | $g_2$ | 4×4 | 8 | 512 | 25 | 208 |
| | $g_3$ | 4×4 | 4 | 256 | 3 | 419 |
| | $g_4$ | 3 | 1 | 64 | 0.1 | 22 |

# 4 An Image Generation Instantiation

In this section, we present a concrete instantiation of our fractal generative model applied to the challenging task of pixel-by-pixel image generation. Although demonstrated here with image data, the underlying divide-and-conquer architecture can potentially extend to other structured, high-dimensional domains. Next, we first discuss the challenges and importance of pixel-by-pixel image generation.

## 4.1 Pixel-by-pixel Image Generation

Pixel-by-pixel image generation remains a significant challenge in generative modeling because of the high dimensionality and complexity of raw image data. This task requires models that can efficiently handle a large number of pixels while effectively learning the rich structural patterns and interdependency between them. As a result, pixel-by-pixel image generation has become a challenging benchmark where most existing methods are still limited to likelihood estimation and fail to generate satisfactory images (Child et al., 2019; Hawthorne et al., 2022; Yu et al., 2023).

Recently, several scale-space autoregressive methods have explored next-scale prediction strategies for image generation (Tian et al., 2024; Tang et al., 2024; Han et al., 2024). However, these methods rely on a single autoregressive model performing full attention across the entire pixel sequence at each scale, resulting in prohibitively high computational complexity for high-resolution images. For example, modeling a 256×256 image at the finest scale using these methods requires each attention matrix in the transformer to have approximately $4 \times 10^9$ elements, making direct modeling of pixels impractical.

Though challenging, pixel-by-pixel generation represents a broader class of important high-dimensional generative problems. These problems aim to generate data element-by-element but differ from long-sequence modeling in that they typically involve non-sequential data. For example, many structures—such as molecular configurations, proteins, and biological neural networks—do not exhibit a sequential architecture yet embody very high-dimensional and structural data distributions. By selecting pixel-by-pixel image generation as an instantiation for our fractal framework, we aim not only to tackle a pivotal challenge in computer vision but also to demonstrate the potential of our fractal approach in addressing the broader problem of modeling high-dimensional non-sequential data with intrinsic structures.

## 4.2   Architecture

As shown in Figure 2, each autoregressive model takes the output from the previous-level generator as its input and produces multiple outputs for the next-level generator. It also takes an image (which can be a patch of the original image), splits it into patches, and embeds them to form the input sequence for a transformer model. These patches are also fed to the corresponding next-level generators. The transformer then takes the output of the previous generator as a separate token, placed before the image tokens. Based on this combined sequence, the transformer produces multiple outputs for the next-level generator.

Following common practices from vision transformers and image generative models (Dosovitskiy et al., 2020; Peebles & Xie, 2023), we set the sequence length of the first generator $g_1$ to 256, dividing the original images into multiple 16×16 patches. The second-level generator then models each patch and further subdivides them into smaller patches, continuing this process recursively. To manage computational costs, we progressively reduce the width and the number of transformer blocks for smaller patches, as modeling smaller patches is generally easier than larger ones. At the final level, we use a very lightweight transformer to model the RGB channels of each pixel autoregressively, and apply a 256-way cross-entropy loss on the prediction. The exact configurations and computational costs for each transformer across different recursive levels and resolutions are detailed in Table 1. Notably, with our fractal design, the computational cost of modeling a 256×256 image is only twice that of modeling a 64×64 image.

Following Li et al. (2024), our method supports different autoregressive designs. In this work, we mainly consider two variants: a raster-order, GPT-like causal transformer (AR) and a random-order, BERT-like bidirectional transformer (MAR) (Figure 6). Both designs follow the autoregressive principle of next-token prediction, each with its own advantages and disadvantages, which we discuss in detail in Appendix B. We name the fractal framework using the AR variant as FractalAR and the MAR variant as FractalMAR.

## 4.3   Relation to Long-Sequence Modeling

Most previous work on pixel-by-pixel generation formulates the problem as long-sequence modeling and leverages methods from language modeling to address it (Child et al., 2019; Roy et al., 2021; Ren et al., 2021; Hawthorne et al., 2022; Yu et al., 2023). However, the intrinsic structures of many data types, including but not limited to images, are beyond one-dimensional sequences. Different from these methods, we treat such data as sets (instead of sequences) composed of multiple elements and employ a divide-and-conquer strategy to recursively model smaller subsets with fewer elements. This approach is motivated by the observation that much of this data exhibits a near-fractal structure: images are composed of sub-images, molecules are composed of sub-molecules, and biological neural networks are composed of sub-networks. Accordingly, generative models designed to handle such data should be composed of submodules that are themselves generative models.

Table 2: **More fractal levels achieve better likelihood estimation performance with lower computational costs**, measured on unconditional ImageNet 64×64 test set. The GFLOPs are measured per forward pass during training with batch size 1. The training time is measured per training iteration on 1 H100 GPU with batch size 8. NLL is reported in bits/dim. The network configurations of $g_1$, $g_2$ and $g_3$ are in Table 1.

|  | Seq Len | | | | | |
|---|---|---|---|---|---|---|
|  | $g_1$ | $g_2$ | $g_3$ | #GFLOPs | Training time (s/iter) | NLL↓ |
| AR, full-length | 64×64×3 | - | - | 29845 | N/A | N/A |
| MAR, full-length | 64×64×3 | - | - | 29845 | N/A | N/A |
| FractalAR (2-level) | 64×64 | 3 | - | 5516 | 4.63 | 3.34 |
| FractalMAR (2-level) | 64×64 | 3 | - | 5516 | 4.63 | 3.36 |
| FractalAR (3-level) | 16×16 | 4×4 | 3 | 438 | 0.28 | 3.14 |
| FractalMAR (3-level) | 16×16 | 4×4 | 3 | 438 | 0.28 | 3.15 |

### 4.4 Implementation

We now describe our fractal generative model implementation for both training and generation processes at a high level. Further details and hyper-parameters can be found in Appendix A. We also provide our source code in the supplementary materials.

**Training.** We train our fractal model end-to-end directly on raw image pixels following a breadth-first manner through the fractal architecture. As shown in Figure 2, during training, each autoregressive module receives outputs from the previous level, processes them through transformer blocks, and generates embeddings for the subsequent fractal level. At the final fractal level, a lightweight autoregressive transformer predicts discrete RGB pixel intensities (from 0 to 255). We compute a cross-entropy loss at this pixel-level prediction and backpropagate it through all fractal levels, optimizing the entire framework in an end-to-end manner.

**Generation.** Our fractal model generates images pixel-by-pixel following a depth-first manner through the fractal hierarchy, as illustrated in Figure 3. Here we use the random-order generation scheme from MAR (Li et al., 2024) as an example. The first-level autoregressive model $g_1$ captures the interdependence between 16×16 image patches in each 256×256 image, and at each step, it produces an embedding for the next level model $g_2$ to generate a patch. Subsequent generators recursively produce outputs to guide the generation of smaller patches. Ultimately, the final-level autoregressive model generates individual pixel RGB values. This recursive generation strategy effectively captures dependencies across multiple image scales, resulting in high-quality generated images.

## 5 Experiments

We conduct extensive experiments on the ImageNet dataset (Deng et al., 2009) with resolutions at 64×64 and 256×256. Our evaluation includes both unconditional and class-conditional image generation, covering various aspects of the model such as likelihood estimation, fidelity, diversity, and generation quality. Accordingly, we report the negative log-likelihood (NLL), Frechet Inception Distance (FID) (Heusel et al., 2017), Inception Score (IS) (Salimans et al., 2016), Precision and Recall (Dhariwal & Nichol, 2021a), and visualization results for a comprehensive assessment of our fractal framework.

### 5.1 Likelihood Estimation

We begin by evaluating our method on unconditional ImageNet 64×64 generation to evaluate its likelihood estimation capability. To demonstrate the effectiveness of our fractal framework, we compare the likelihood estimation performance of our framework with different numbers of fractal levels, as shown in Table 2. Modeling the entire 64×64×3=12,288-pixel sequence with a single autoregressive model incurs prohibitive computational costs and makes training infeasible. Moreover, a two-level fractal framework that first models the entire pixel sequence and then the RGB channels requires over ten times the computation of our three-

Table 3: Comparison with other likelihood-based models on unconditional ImageNet 64×64 test set. NLL is reported in bits/dim.

| | type | NLL↓ |
|---|---|---|
| iDDPM (Nichol & Dhariwal, 2021) | diffusion | 3.53 |
| FM (Lipman et al., 2022) | diffusion | 3.31 |
| NFDM (Bartosh et al., 2024) | diffusion | 3.20 |
| Wavelet Flow (Yu et al., 2020) | norm. flow | 3.78 |
| TarFlow (Zhai et al., 2024) | norm. flow | 2.99 |
| PixelCNN (van den Oord et al., 2016) | AR | 3.57 |
| Combiner AR (Ren et al., 2021) | AR | 3.42 |
| Perceiver AR (Hawthorne et al., 2022) | AR | 3.40 |
| MegaByte (Yu et al., 2023) | AR | 3.40 |
| **FractalAR** | fractal | 3.14 |
| **FractalMAR** | fractal | 3.15 |

Table 4: Sample quality evaluation on ImageNet 256×256 pixel-level generation. FractalMAR is the only **pixel-by-pixel** framework in this table. We also provide the performance of other pixel-level generation methods for reference.

| | type | #params | FID↓ | IS↑ | Pre.↑ | Rec.↑ |
|---|---|---|---|---|---|---|
| BigGAN-deep (Brock et al., 2018) | GAN | 160M | 6.95 | 198.2 | **0.87** | 0.28 |
| GigaGAN (Kang et al., 2023) | GAN | 569M | 3.45 | 225.5 | 0.84 | **0.61** |
| StyleGAN-XL (Sauer et al., 2022) | GAN | 166M | 2.30 | 265.1 | 0.78 | 0.53 |
| ADM Dhariwal & Nichol (2021b) | diffusion | 554M | 4.59 | 186.7 | 0.82 | 0.52 |
| Simple diffusion Hoogeboom et al. (2023) | diffusion | 2B | 3.54 | 205.3 | - | - |
| VDM++ Kingma & Gao (2023) | diffusion | 2B | 2.12 | 267.7 | - | - |
| SiD2 Hoogeboom et al. (2024) | diffusion | - | **1.38** | - | - | - |
| JetFormer Tschannen et al. (2024) | AR+flow | 2.8B | 6.64 | - | 0.69 | 0.56 |
| **FractalMAR-B** | fractal | 186M | 11.80 | 274.3 | 0.78 | 0.29 |
| **FractalMAR-L** | fractal | 438M | 7.30 | 334.9 | 0.79 | 0.44 |
| **FractalMAR-H** | fractal | 848M | 6.15 | **348.9** | 0.81 | 0.46 |

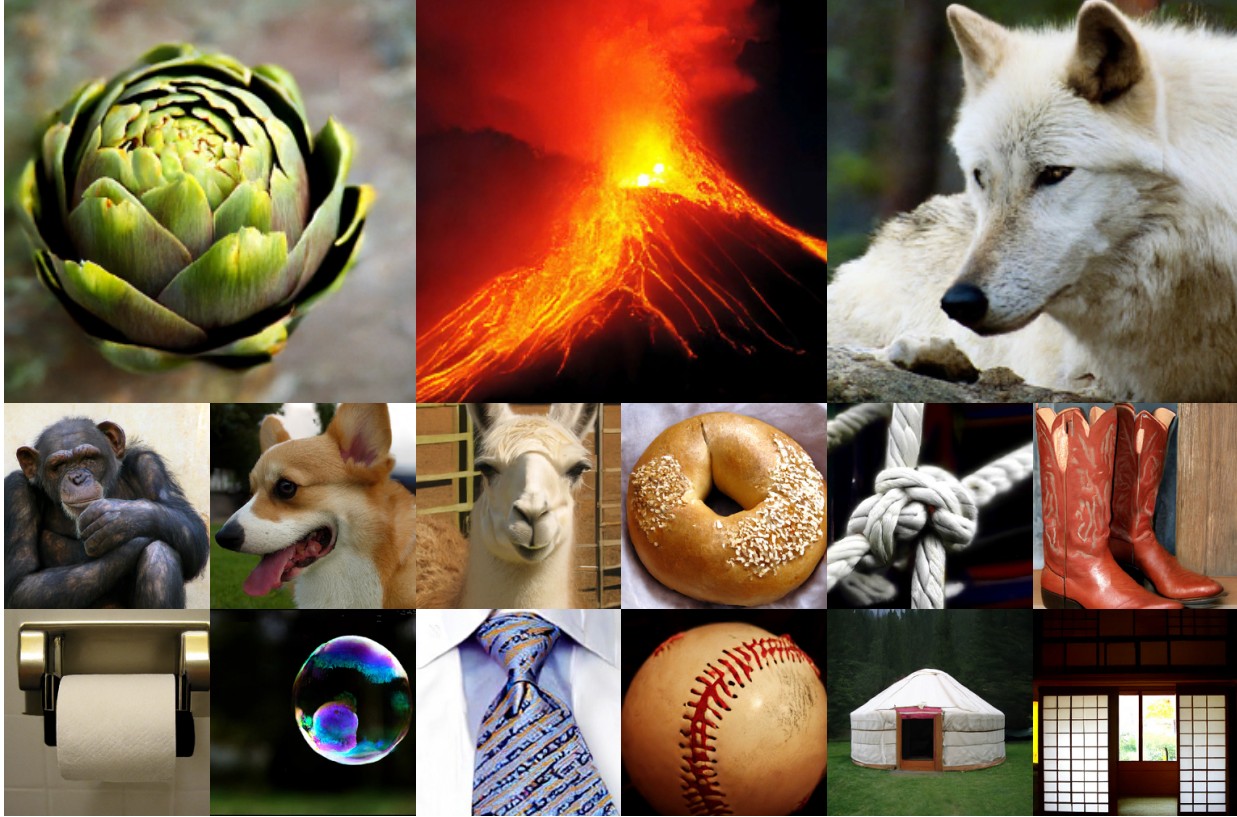

Figure 4: **Pixel-by-pixel generation results from FractalMAR-H on ImageNet 256×256.** Our fractal method can generate high-quality high-resolution images in a pixel-by-pixel manner, with an average throughput of 1.29 seconds per image. More qualitative results are in Figure 7.

level fractal model. Employing more fractal levels is not only more computationally efficient but also improves likelihood estimation performance, likely because it better captures the intrinsic hierarchical structure of images.

We further compare our method with other likelihood-based models in Table 5. Our fractal generative model, instantiated with both causal and masked autoregressive fractal generators, achieves strong likelihood performance. In particular, it achieves a negative log-likelihood of 3.14 bits per dim, which outperforms the previous best autoregressive model (3.40 bits per dim) by a significant margin and remains competitive with advanced diffusion-based and normalizing-flow-based methods. These findings demonstrate the effectiveness

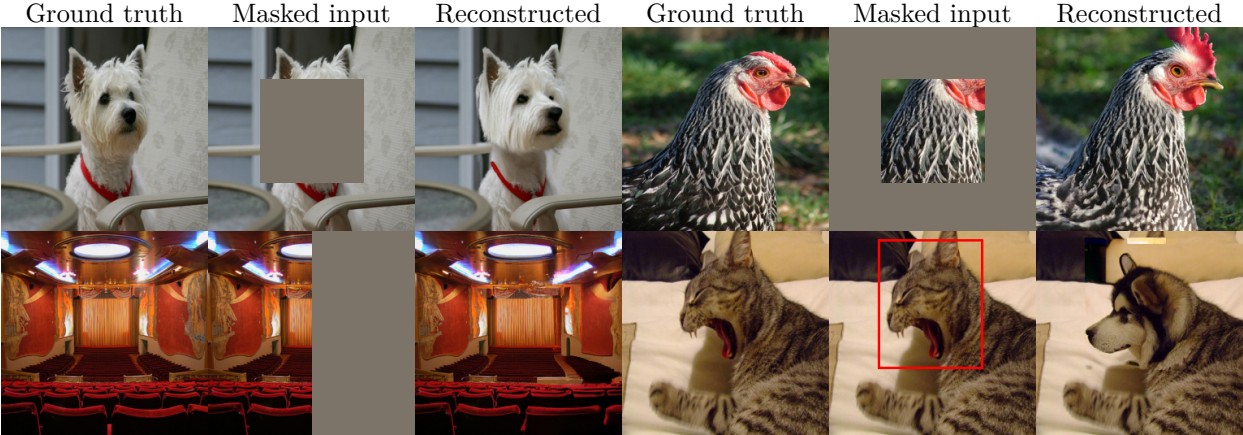

Figure 5: Conditional pixel-by-pixel prediction results, including image inpainting (first row left half), outpainting (second row left half), uncropping (outpainting on a large mask, first row right half), and class-conditional editing (inpainting with another class label, second row right half).

of our fractal framework on the challenging task of pixel-by-pixel image generation, highlighting its promising potential in modeling high-dimensional non-sequential data distributions.

## 5.2 Generation Quality

We also evaluate FractalMAR on the challenging task of class-conditional image generation at a resolution of 256×256, using four fractal levels. We report standard metrics including FID, Inception Score, Precision, and Recall following standard practices to evaluate its generation quality in Table 4. Specifically, FractalMAR-H achieves an FID of 6.15 and an Inception Score of 348.9, with an average throughput of 1.29 seconds per image (evaluated at a batch size of 1,024 on a single Nvidia H100 PCIe GPU). Notably, our method achieves strong Inception Score and Precision, indicating its ability to generate images with high fidelity and fine-grained details, as also demonstrated in Figure 4. However, its FID and Recall are relatively weaker, suggesting that the generated samples are less diverse compared to other methods. We hypothesize that this is due to the significant challenge of modeling nearly 200,000 pixels in a pixel-by-pixel way. Nonetheless, these results highlight our method's effectiveness in not only accurate likelihood estimation but also the generation of high-quality images.

We further observe a promising scaling trend: increasing the model size from 186M to 848M parameters substantially improves the FID from 11.80 to 6.15 and the Recall from 0.29 to 0.46. We expect that additional scaling of parameters could narrow the gap in FID and Recall even further. Unlike models that rely on tokenizers, our method is free from the reconstruction errors introduced by tokenization, suggesting the potential for uncapped performance gains with larger model capacities.

## 5.3 Conditional Pixel-by-pixel Prediction

We further examine the conditional pixel-by-pixel prediction performance of our method using conventional tasks in image editing. Figure 5 provides several examples, including inpainting, outpainting, uncropping, and class-conditional editing. As shown in the figure, our method can accurately predict the masked pixels based on the unmasked regions. Additionally, it can effectively capture high-level semantics from class labels and reflect it in the predicted pixels. This is illustrated in the class-conditional editing example, where the model replaces a cat's face with that of a dog by conditioning on the dog's class label. These results demonstrate the effectiveness of our method in predicting unknown data given known conditions.

More broadly, by generating data element-by-element, our method provides a generation process that is more interpretable for humans compared to approaches such as diffusion models or generative models operating in latent spaces. This interpretable generation process not only allows us to better understand how data is generated but also offers a way to control and interact with the generation. Such abilities are particularly

important in applications like visual content creation, architectural design, and drug discovery. Our promising results highlight the potential of our approach for controllable and interactive generation, paving the way for future explorations in this direction.

## 6 Discussion and Conclusion

The effectiveness of our proposed fractal generative model, demonstrated through the challenging task of pixel-by-pixel generation, suggests new opportunities for designing generative models. It highlights the potential of dividing complex data distributions into manageable sub-problems and addressing them by using existing generative models as modular units. We believe that fractal generative models are particularly suited for modeling data with intrinsic structures beyond sequential orders. We hope that the simplicity and effectiveness of our method will inspire the research community to explore novel designs and applications of fractal generative models.

**Broader Impacts.** Our primary aim is to advance the fundamental research on generative models. Similar to all generative models, there are potential negative societal consequences if our approach is misused to produce disinformation or amplify biases. However, these considerations lie outside the scope of this paper and are therefore not discussed in detail.

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

# A    Implementation Details

Here we provide more step-by-step implementation details of the training and generation process of our pixel-by-pixel image generation instantiation. We also include the codes to reproduce our results in the supplementary materials. All codes and models will be made publicly available.

**Training.** Below is a detailed step-by-step description of the training procedure on 256×256 images:

- At the highest fractal level, generator $g_1$ receives the class embedding (or condition embedding) as input, along with embeddings of 16×16 image patches. The transformer within $g_1$ processes these embeddings to produce output embeddings that serve as conditional inputs for the next fractal level $g_2$. Due to the autoregressive nature of the generator, the training can be done in a teacher-forcing way that produces embeddings and receives gradients from all patches in one training iteration.
- The second-level generator $g_2$ then takes these conditional embeddings from $g_1$ and combines them with embeddings of 4×4 patches in each 16×16 patch. The transformer of $g_2$ processes these combined inputs and generates further refined embeddings to be used as conditional inputs for the subsequent fractal level $g_3$.
- Similarly, the third-level generator $g_3$ combines the conditional embeddings from $g_2$ with the embeddings of all pixels in each 4×4 patch. Its transformer outputs embeddings that condition the final fractal level $g_4$.
- At the final fractal level $g_4$, a specialized pixel-level autoregressive transformer predicts discrete RGB values for individual pixels. These pixel-level RGB intensities are discretized into 256 levels, and a cross-entropy loss is computed between these predictions and the ground-truth pixel values from the input images.
- Finally, the computed cross-entropy loss at the pixel level is backpropagated through all fractal levels and trains the entire framework in an end-to-end manner.

The backward pass is simply the automatic differentiation of the forward pass. In summary, each fractal generator receives conditioning signals from the previous level and produces new conditions for the next. The final level computes the cross-entropy loss, and gradients are propagated in reverse order.

When modeling high-resolution images with multiple levels of autoregressive models, we find it slightly helpful to include a "guiding pixel" in the autoregressive sequence. Specifically, the model first uses the output from the previous generator to predict the average pixel value of the current input image. This average pixel value then serves as an additional condition for the transformer. In this way, each generator begins with a global context before predicting fine-grained details. We apply this "guiding pixel" only in experiments on ImageNet 256×256.

Since our autoregressive model divides images into patches, it can lead to inconsistent patch boundaries during generation. To address this issue, we provide the next-level generator not only the output at the current patch but also the outputs at the surrounding four patches. Although incorporating these additional inputs slightly increases the sequence length, it significantly reduces the patch boundary artifacts.

By default, the models are trained using the AdamW optimizer (Loshchilov & Hutter, 2019) for 800 epochs (the FractalMAR-H model is trained for 600 epochs). The weight decay and momenta for AdamW are 0.05 and (0.9, 0.95). We use a batch size of 2048 for ImageNet 64×64 and 1024 for ImageNet 256×256, and a base learning rate (lr) of 5e-5 (scaled by batch size divided by 256). The model is trained with 40 epochs linear lr warmup (Goyal et al., 2017), followed by a cosine lr schedule. The 64×64 model takes 3.5 days, and the 256×256 FractalMAR-L model takes 7.6 days on 32 H100 GPUs.

**NLL computation.** Since our method uses a divide-and-conquer approach by recursively applying autoregressive decompositions, the computation of the NLL effectively becomes recursively computing the NLL for each autoregressive model. Specifically, the NLL of an image is computed as

$$-\log p(x_1, \cdots, x_{k^n}) = -\sum_{i=1}^{k} \log p(x_{(i-1)\cdot k^{n-1}+1}, \cdots, x_{i\cdot k^{n-1}} | x_1, \cdots, x_{(i-1)\cdot k^{n-1}}).$$

This means the overall image NLL is simply the summation of individual NLL values on each patch conditioned on previous patches. Furthermore, the NLL of each individual patch itself is also recursively computed as the sum of NLL values of the smaller patches in it, and so on. The NLL on each pixel channel is computed w.r.t. the ground-truth pixel value using cross-entropy on 256 discrete values.

**Generation.** Here we describe the generation process step-by-step, using the generation of a $256{\times}256$ image as an example. To provide additional clarity, we also include visualization videos illustrating this recursive generation process in the supplementary materials.

- The first-level generator $g_1$ models the interdependence between the $16{\times}16$ patches in an $256{\times}256$ image. At each step of $g_1$, it uses its transformer to produce an embedding for a $16{\times}16$ patch to generate, conditioned on the previously generated patches. Then, it asks the second-level generator $g_2$ to generate this patch based on the embedding it produced.
- The second-level generator $g_2$ operates similarly but at a finer scale. It models the interdependence between the $4{\times}4$ patches in a $16{\times}16$ image. Specifically, at each step of $g_2$, it uses its transformer to produce an embedding for the next $4{\times}4$ patch to generate, conditioned on previously generated $4{\times}4$ patches and the output embedding from $g_1$. Then, it asks the third-level generator $g_3$ to generate this $4{\times}4$ patch based on this embedding.
- The third-level generator $g_3$ models and generates individual pixels within each $4{\times}4$ patch. For each step of $g_3$, it employs its transformer to generate embeddings for the next pixel to generate, conditioned on the previously generated pixels in the patch and the output embedding from $g_2$. It then asks the final-level generator $g_4$ for the actual RGB value generation.
- At the final fractal level, the generator $g_4$ samples RGB values for each pixel autoregressively from probability distributions produced by its transformer's output logits. These RGB predictions represent the final pixel values of the generated image.

Following common practices in the literature (Chang et al., 2022; 2023; Li et al., 2023), we implement classifier-free guidance (CFG) and temperature scaling for class-conditional generation. To facilitate CFG (Ho & Salimans, 2022), during training, the class condition is replaced with a dummy class token in 10% of the samples. At inference time, the model is run with both the given class token and the dummy token, yielding two sets of logits $l_c$ and $l_u$ for each pixel channel. The predicted logits are then adjusted according to the following equation: $l = l_u + \omega \cdot (l_c - l_u)$, where $\omega$ is the guidance scale. We employ a linear CFG schedule during inference for the first-level autoregressive model, as described in Chang et al. (2023). We perform a sweep over different combinations of guidance scale and temperature to identify the optimal settings for each conditional generation model.

We also observed that CFG can suffer from numerical instability when the predicted probability for a pixel value is very small. To mitigate this issue, we apply top-p sampling with a threshold of 0.0001 on the conditional logits before applying CFG.

**Fractal Factorization.** We provide a concrete walkthrough of the recursive factorization used in our fractal autoregressive framework. We aim to model a joint distribution over $N$ random variables, $p(x_1, x_2, \ldots, x_N)$. Directly factorizing this distribution with a single autoregressive model becomes impractical for large $N$. We therefore adopt a divide-and-conquer strategy: we fix a block size $k$ and choose the number of recursive levels $n$ such that $N = k^n, n = \log_k N$. At each level, an autoregressive module models a conditional probability of the form $p(x|c)$, where $x$ denotes a short sequence of length $k$ and $c$ is the context produced by the previous level.

Now, we provide a two-level example when $N = k^2$. We partition the variables into $k$ consecutive groups $G_i = \{x_{(i-1)k+1}, \ldots, x_{ik}\}, i = 1, \ldots, k$. By the chain rule,

$$p(x_1, \ldots, x_{k^2}) \;=\; \prod_{i=1}^{k} p(G_i \,|\, G_1, \ldots, G_{i-1}). \tag{1}$$

This conditional probability distribution *between* groups is modeled by the level-1 autoregressive model. Then, each term on the right is a distribution over $k$ variables, which we further factorize with a *lower-level*

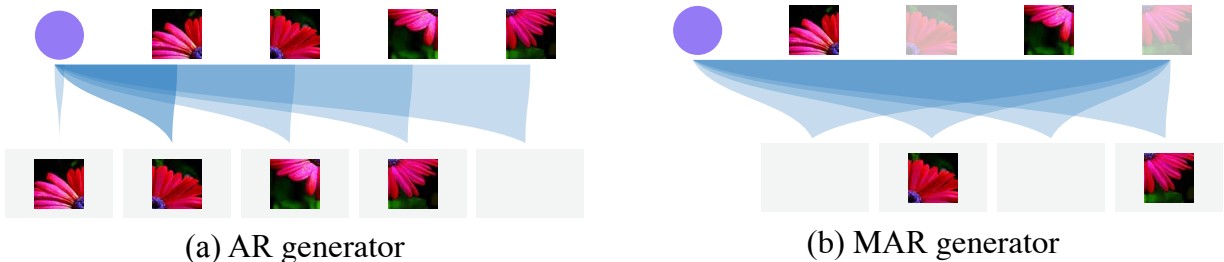

(a) AR generator                         (b) MAR generator

Figure 6: Two variants for autoregressive modeling. The AR variant models the sequence in a raster-scan order using a causal transformer. The MAR variant models the sequence in a random order using a bidirectional transformer: at each generation step, the model randomly selects masked positions to fill in, resulting in a random sampling order. Both are valid generators to build our fractal framework.

AR module:

$$p(G_i \,|\, G_1, \ldots, G_{i-1}) \;=\; \prod_{j=1}^{k} p\big(x_{(i-1)k+j} \,\big|\, G_1, \ldots, G_{i-1}, \, x_{(i-1)k+1}, \ldots, x_{(i-1)k+j-1}\big). \tag{2}$$

This conditional probability distribution *within* each group is modeled by the level-2 autoregressive model.

Now, let $N = k^n$. The top level groups the sequence into $k$ contiguous blocks of size $k^{n-1}$:

$$p(x_1, \cdots, x_{k^n}) = \prod_{i=1}^{k} p(x_{(i-1)\cdot k^{n-1}+1}, \cdots, x_{i\cdot k^{n-1}} | x_1, \cdots, x_{(i-1)\cdot k^{n-1}}).$$

Each conditional on the right is again a joint distribution over $k^{n-1}$ variables, which is factorized *by the next fractal level* using the same rule, now with exponent $n-1$. After $n$ applications of this rule, we reach modules that operate on sequences of length $k$, completing the recursive factorization.

## B   Additional Results

Table 5: Comparison on conditional ImageNet 64×64 with classical generative models.

|  | type | FID↓ |
|---|---|---|
| iDDPM (Nichol & Dhariwal, 2021) | diffusion | 2.92 |
| StyleGAN-XL (Sauer et al., 2022) | GAN | **1.51** |
| Consistency Model (Song et al., 2023) | consistency | 4.70 |
| MAR (Li et al., 2024) | AR+diffusion | 2.93 |
| **FractalAR** | fractal | 6.30 |
| **FractalMAR** | fractal | 2.76 |

**Class-conditional ImageNet 64×64.** We evaluate class-conditional image generation on ImageNet 64×64, reporting FID according to standard practice. Consistent with PixelCNN (van den Oord et al., 2016), we find that class conditioning has negligible impact on NLL but substantially improves visual quality and FID. The results demonstrate that our fractal generative model can achieve competitive performance as classical generative models.

We also compare the performance of the AR and MAR variants, whose structures are illustrated in Figure 6. The AR variant leverages key-value caching to accelerate generation, whereas the MAR variant employs bidirectional attention, which aligns more naturally with image modeling and enables the parallel prediction of multiple patches, thereby improving computational efficiency. As shown in the table, both of our models achieve favorable performance, with FractalMAR outperforming FractalAR overall, as also demonstrated in Li et al. (2024). Therefore, we choose to use the MAR variant for conditional image synthesis on ImageNet at a resolution of 256×256.

Table 6: Different orders of channel modeling within each pixel can slightly influence generation performance. All models are trained for 400 epochs on ImageNet 64×64.

| order | NLL↓ | FID↓ |
|---|---|---|
| Y→Cb→Cr | N/A | 3.55 |
| B→R→G | 3.17 | 3.32 |
| G→R→B | 3.17 | 3.14 |
| R→G→B | 3.17 | 3.15 |

**Modeling Pixels.** We also examine how different pixel modeling orders affect performance. We experimented with three autoregressive orders: RGB, GRB, and BGR, as well as converting the RGB channels to the YCbCr color space. The results are summarized in Table 6. We found that while all orders achieved similar negative log-likelihood values, the FID varied slightly among the autoregressive orders (note that the NLL in the YCbCr space is not comparable with that in the RGB space). This variation is likely because, akin to human vision, the Inception model used to compute FID places greater emphasis on the red and green channels than on the blue channels (Mustafi et al., 2009). Nonetheless, the choice of autoregressive order does not result in significant performance differences, demonstrating the robustness of our method.

**Additional qualitative results.** We further provide additional conditional generation results on ImageNet 256×256 in Figure 7, additional conditional pixel-by-pixel prediction results in Figure 8, and also include videos that demonstrate our generation process in the supplementary material.

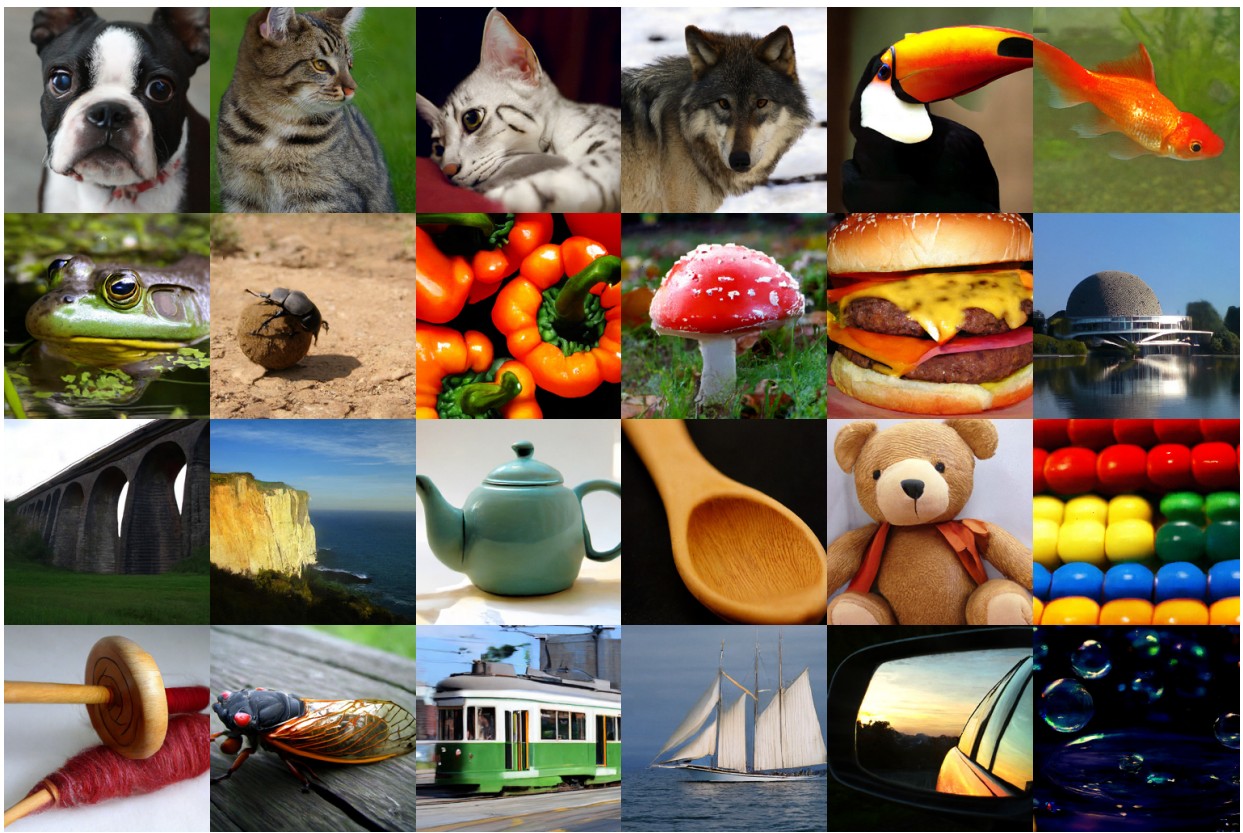

Figure 7: Additional pixel-by-pixel generation results from FractalMAR-H on ImageNet 256×256.

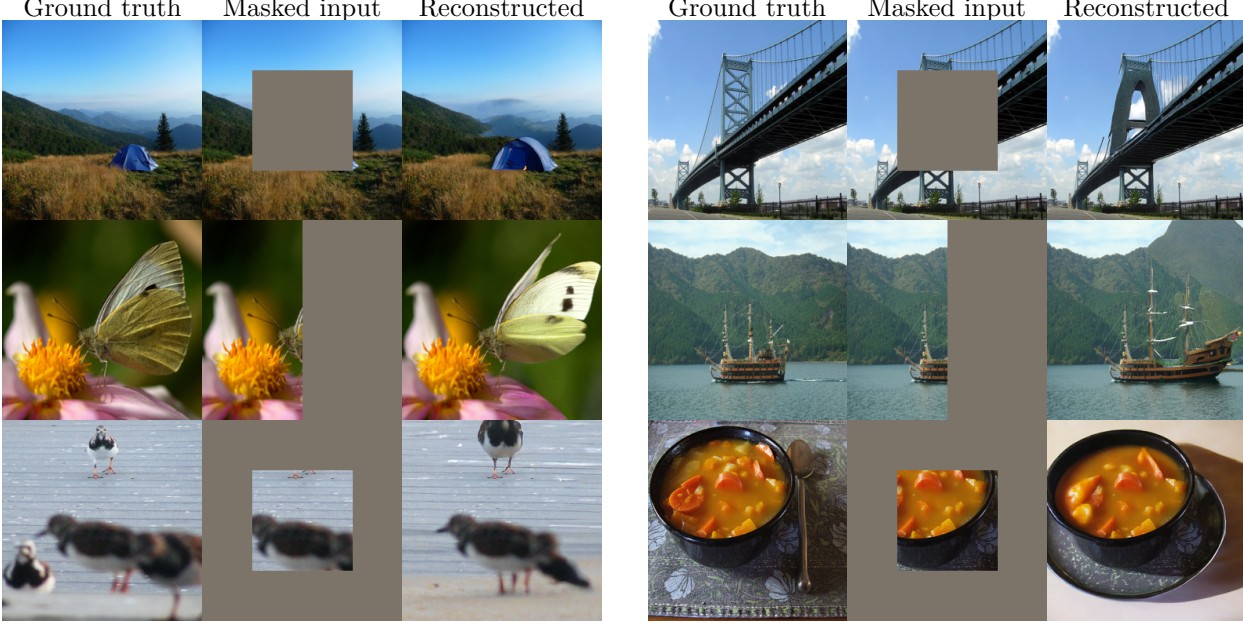

Figure 8: Additional conditional pixel-by-pixel prediction results, including image inpainting, outpainting, and uncropping.

