# OpenReview forum: "Fractal Generative Models"
_TMLR — Accepted by TMLR_

### Review · Reviewer_JpUc · 2025-10-11

**Summary Of Contributions:**

Inspired by the concept of fractals, this paper proposes a new type of generative models, called fractal geenrative models. The fractal generative model generates high-dimensional data (e.g., images) by recursively calling layers—instantiated as autoregressive models. The intuition is that layers of autoregressive model leads to a more expressive model. This paper validates the effectiveness of fractal geenrative models on the task of pixel-by-pixel image generation.

**Additional Comments:**

1. The key idea of the proposed method is formulated as the formula shown in Section 3.2. The equality in this formula holds **when** $x_1 \ldots x_{(i-1)k^{n-1}}$ are the parents of $x_{(i-1)k^{n-1}} \ldots x_{ik^{n-1}}$ in the underlying causal structure, which does not generally hold in the task of pixel-by-pixel image generation. So, I assume the RHS is simply an approximation of the joint probability on LHS. From this perspective, a single autoregressive model directly modeling the joint probability could lead to better performance compared with using fractal generative models. My question is: is there any result comparing their performance? and is there any result comparing their computational complexity?
2. How to determine the number of recursive levels in experiments? Is there any ablation studies on this?
3. The authors propose a fractal generative framework that could work in various scenarios. I just wonder if it is possible to apply this framework to 1) diffusion-model-based generators, 2) language generation tasks?
4. Is there intuition why generating non-sequential data in a sequential manner may lead to good performance?

**Audience:**

Yes

**Audience Explanation:**

This paper would be of interest to researchers from the field of generative models.

**Broader Impact Concerns:**

None.

**Claims And Evidence:**

Yes

**Claims Explanation:**

They provide the generated images in Figure 4, and compare with other types of generative models in Table 2,3.

**Requested Changes:**

Elaborate on the differences with MAR in the related work section.

---

### Review · Reviewer_5J1v · 2025-10-28

**Summary Of Contributions:**

This paper proposed a new paradigm of modularization method, to abstract generative models into atomic models, involving a new class of models called fractal generative models. Fractal generative models recursively invoke atomic generative models, resulting in fractal self-similar structures. Numerical experiment of the proposed fractal framework on autoregressive model used on pixel-by-pixel image generation shows strong performance in both likelihood estimation and generation quality.

**Additional Comments:**

This paper has great contribution and theoretical novelty, potentially preceding new lines of future researches. The numerical results are also good. Many implementation details are also provided. Therefore, I recommend accept.

**Audience:**

Yes

**Audience Explanation:**

This paper has great contribution and theoretical novelty, potentially preceding new lines of future researches.

**Claims And Evidence:**

Yes

**Claims Explanation:**

Numerical evidence is solid. Given the extent of great contributions in all other dimensions, the lacking of theoretical proofs is overall acceptable.

**Requested Changes:**

No mandatory change is required. With that being said, the reviewer would prefer much more theories with rigorous proofs, in addition to the short theory provided in section 3.2.

---

> ### Author Response · Authors · 2025-11-16
>
> We’ve uploaded a revised manuscript, and the factorization equation is now detailed in the final subsection of Appendix A.

---

### Review · Reviewer_JX8i · 2025-11-10

**Summary Of Contributions:**

The authors aim to establish a new paradigm with fractal generative models. The core idea is to recursively instantiate a generative model, thereby modeling a sequence problem across several hierarchical levels. This is achieved by recursively decomposing the joint distribution, which is purported to not only reduce computational costs compared to a single large autoregressive model but also to capture the intrinsic hierarchical structure within the data.

The biggest strength of this submission are in its novelty and strong results. The method is instantiated using autoregressive modules and evaluated on the challenging task of pixel-by-pixel image generation on ImageNet. On the 64x64 dataset, the model achieves strong likelihood estimation, reporting a negative log-likelihood of 3.14 bits/dim, which surpasses previous autoregressive models. On the 256x256 dataset, the model demonstrates high-quality, class-conditional generation, achieving an FID of 6.15 and an Inception Score of 348.9. The paper also provides qualitative results for conditional prediction tasks such as image inpainting, outpainting, and class-conditional editing.

The major weaknesses are linked to the evaluation protocol, implementation details, and presentation, as I mention in the Requested Changes below.

**Audience:**

Yes

**Audience Explanation:**

The paper addresses the significant challenge of generative modeling for high-dimensional, structured data, a problem of clear and persistent interest to the TMLR community. The work's strengths, which will attract audience interest, are numerous: conceptual novelty, computational efficiency, strong results, and promising potential. On the latter point, while tested on images, the fractal framework is conceptually general. It holds clear potential for other data modalities that exhibit intrinsic hierarchical or self-similar structures (e.g., molecular data, proteins, time-series), which will interest a broad segment of the community.

**Broader Impact Concerns:**

The authors did not include a Broader Impact Statement, even though powerful image generators like the proposed can significantly cause societal harm in terms of surveillance, deepfakes, etc.

**Claims And Evidence:**

Yes

**Claims Explanation:**

Yes, the claims are largely supported by the provided evidence. The experiments section, for instance, details the image completion and generation tasks in sufficient detail, providing clear evidence for the model's capabilities in this specific domain. The quantitative results presented in Tables 2, 3, and 4 are strong, demonstrating state-of-the-art performance in likelihood estimation for pixel-by-pixel models and highly competitive generation quality. The ablation in Table 2, which shows improved performance and lower computational cost with more fractal levels, provides direct support for the core "fractal" hypothesis. Furthermore, the qualitative results in Figures 4, 5, 7, and 8 are convincing, showing high-fidelity generation and plausible conditional edits, which visually substantiates the numerical claims.

**Requested Changes:**

I think this is extremely strong work, but the following changes are suggested to strengthen it significantly. A few of them are critical for clarifying the method's novelty, correctness, and generalizability.

1. _(Critical) Unaddressed Autodiff Implementation Challenges:_ A critical, unaddressed implementation challenge relates to automatic differentiation. The paper must include a detailed section discussing this. As formulated, a recursive, end-to-end trained model with $L$ levels would seem to require $L$-th order derivatives (e.g., Hessians for two levels, 3rd-order derivatives for three) to backpropagate gradients from the innermost loss to the outermost generator's parameters. This is computationally prohibitive and a potential flaw in the proposed training scheme. The authors must clarify how this is managed. If my analysis is incorrect, they must explain _why_ and provide a precise, technical description of the gradient flow.

2. _(Clarification) Missing Citations and Positioning:_ The related work section is missing critical citations. The authors should include and discuss the following, as they appear highly relevant:

    - "Hierarchical Reasoning Model", Wang et al., arXiv 2025. This paper's use of two levels of modeling (high-level and low-level) seems directly analogous to the fractal concept and should be addressed.

    - "Weight-Space Linear Recurrent Neural Networks", Nzoyem et al., arXiv 2025. The authors should differentiate their contribution from these works.

3. _(Clarification) Limited Testbeds and Task-Specific Justification:_ The reliance on pixel-by-pixel image generation as the sole testbed is a significant limitation. While an interesting example, its challenges are primarily dominated by the quadratic scaling at high resolutions. The authors should clarify that this task is not intrinsically challenging at low resolutions (e.g., 64x64). To substantiate the claims of generality, the method should be evaluated on additional testbeds, preferably non-image data, to demonstrate that the fractal approach is more than just an efficient architecture for images.

4. _(Critical) Lack of Ablation Studies:_ While I appreciate Table 2, I also think that the paper lacks crucial ablation studies. To confirm the method's potential, the authors must include an ablation on data _without_ a clear intrinsic hierarchical structure. This would validate whether the model's benefits are due to its fractal nature (as claimed) or if it is simply a well-performing hierarchical architecture in general.

5. _(Clarification) Parameter Scaling:_ The paper states, "Each fractal generator level uses independently parameterized modules," which implies the total parameter count grows with depth. The only mitigation mentioned is reducing the width of inner Transformer blocks. This seems insufficient. The authors must provide a more thorough analysis of parameter scaling with depth and justify why this is not a significant drawback. How does it work with generative models different from Transformers?

6. _(Critical) Core Factorization Equation:_ The core factorization equation in Section 3.2 ("The joint distribution factorizes recursively as...") requires significant clarification. The notation is dense and not immediately obvious. The authors must provide a simpler, concrete example (e.g., for $n=2$ levels) to walk the reader through how the joint distribution is recursively decomposed.

7. _(Clarification) Loss Function:_ The authors should elaborate on the final loss computation.

    1. How is the 256-way cross-entropy loss applied across three independent RGB channels (is it a $3 \times 256$ problem, or just $256$)?

    2. They must justify the choice of a cross-entropy loss over a simpler, and perhaps more perceptually relevant, MSE loss in RGB space, especially given that the data is continuous-valued before discretization.

8. _(Clarification) Figure 6:_ Figure 6(b) (MAR) is confusing. It is meant to illustrate a "random order" but still depicts patches in what appears to be a standard raster-scan order. The authors must clarify what "random order" means in this context or remake the figure to accurately reflect the MAR sampling process.

9. _(Clarification) Figure 1:_ The caption for Figure 1 states "Four levels shown" but visually, only three distinct generator levels appear to be depicted. This discrepancy should be corrected for clarity.

10. _(Clarification) Table 2:_ The naming in Table 2 is ambiguous. The third and fourth rows ("AR, 2-level", "MAR, 2-level") are not clearly differentiated from the "FractalAR (3-level)" and "FractalMAR (3-level)" models. Are the 2-level models not also "FractalAR/MAR"? The authors must clarify the precise architectures being compared in this table.

---

> ### Author Response · Authors · 2025-11-16
>
> We thank the reviewer for the positive comments and insightful suggestions to help us improve the clarity and completeness of the paper. Below, we address each major concern in detail.
>
> ***Autodiff Implementation***
>
> Our implementation **does not require higher-order derivatives**. As detailed in Figure 2 and Appendix A, the backward pass is simply the automatic differentiation of the forward pass. Each fractal generator receives conditioning signals from the previous level and produces new conditions for the next. The final level computes the cross-entropy loss, and gradients are propagated in reverse order–standard first-order backpropagation. No Hessians or higher-order derivatives are involved. We have clarified this more explicitly in the revised Appendix A.
>
> ***Missing Citations and Positioning***
>
> We appreciate the reviewer for pointing out the relevant works and have included them in the revised paper. However, our method is conceptually distinct. The first paper uses a two-level hierarchy separating high-level planning from low-level execution, whereas our model employs recursive autoregressive composition to build deeper hierarchies of generative models. The second paper parameterizes RNN hidden states as the weights of another neural network. In contrast, our approach focuses on recursive probabilistic factorization, stacking autoregressive generators to capture multi-scale dependencies.
>
> ***Additional Testbeds and Ablations***
>
> Due to the time constraints of the rebuttal phase, we are unable to include additional results on non-image data. Nonetheless, as emphasized in the final two paragraphs of the introduction, our design specifically targets **non-sequential data with intrinsic hierarchical structures** and uses images as a representative and challenging testbed of such data. For data lacking such structure (e.g., Gaussian noise), we do not expect the fractal model to show clear advantages.
>
> ***Parameter Scaling***
>
> While the number of parameters increases with the number of fractal levels, the number of tokens our model can handle grows exponentially. Therefore, the required depth increases only logarithmically with sequence length, keeping total parameters modest. Moreover, we observe that in practice, different fractal levels require different capacities: allocating more parameters to early (high-level) generators and fewer to later (fine-level) ones yields both efficiency and strong performance (see Table 1). Intuitively, it is much harder to model cross-patch dependencies in an image than cross-pixel dependencies in a small patch (e.g., 4x4), so earlier levels need greater capacity. We also present a parameter scaling analysis in Table 4, where models of 186M, 438M, and 848M parameters consistently improve, confirming stable and efficient scalability.
>
> ***Factorization equation***
>
> We thank the reviewer for the suggestion. We have provided a detailed example of the fractal factorization in the revised Appendix A.
>
> ***Loss function***
>
> As described in Appendix A, a specialized pixel-level autoregressive transformer sequentially predicts the three color channels (RGB). Each channel has 256 discrete values, and we apply a 256-way cross-entropy loss for each. Using MSE loss would make the model **deterministic and non-samplable**, which is against its generative nature. The cross-entropy loss preserves stochasticity and enables diverse sampling, which is essential for generative modeling.
>
> ***MAR figure***
>
> We will clarify Figure 6 and its caption. In MAR, at each step, the model **randomly selects masked positions** to fill in, resulting in a random sampling order. The previous figure’s layout may have appeared raster-like, but the underlying generation order is random and has been explicitly described in the revised caption.
>
> ***Teaser figure***
>
> Figure 1 indeed depicts **four fractal levels**, represented by blue, yellow, purple, and green modules. The caption clarifies this, noting that finer structures is “better viewed zoomed in”, consistent with the “fractal” concept of self-similarity across scales.
>
> ***Table 2***
>
> We have changed the naming to 2-level fractal AR/MAR to prevent misunderstanding.
>
> ***Broader impact***
>
> We thank the reviewer for pointing this out. We have included a broader impact statement at the end of the revised main paper.

---

> > ### Comment · Reviewer_JX8i · 2025-11-16
> >
> > Dear Authors,
> >
> > Thank you for your reply. I understand your paper much better now, and I can appreciate its beauty even more.
> > As I indicated in my original review, I think this is strong work, and your reply has confirmed this. My apologies for not zooming long enough on Figure 1 (the smallest fourth level might require a colour with more presence).
> >
> > Is it possible on your end to update the manuscript so that I can check how you addressed other requested changes (e.g. the factorisation equation)?
> >
> > Thank you,

---

> > > ### Author Response · Authors · 2025-11-16
> > >
> > > Dear reviewer,
> > >
> > > Thank you for the thoughtful follow-up and kind words. We’ve uploaded a revised manuscript reflecting the requested changes. The factorization equation is now detailed in the final subsection of Appendix A.

---

> > > > ### Comment · Reviewer_JX8i · 2025-11-16
> > > >
> > > > Received, thank you! All my concerns appear to have been resolved.

---

### Decision · Action_Editor_dx5M · 2025-12-19

**Recommendation:** Accept as is

**Audience:**

Yes

**Audience Explanation:**

All the reviewers agree this contribution is of interest to the TMLR community interested in generative models.

**Claims And Evidence:**

Yes

**Claims Explanation:**

This paper introduces fractal generative models, a new paradigm that recursively composes generative models as modular building blocks, yielding self-similar architectures. The key idea is to decompose the joint distribution of the model into manageable subproblems, reducing computation while capturing hierarchical structure. The authors demonstrate their model achieve state-of-the-art performance in ImageNet generation.

Overall, the reviewers were enthusiastic with the idea proposed in this work. Some concerns were raised concerning the correctness and feasibility of the recursive training procedure (Reviewer JX8i), as well as the theoretical basis for the model (Reviewer JpUc). Most of these concerns seemed to have been resolved during the rebuttal and discussion.

For this reason, I recommend this paper for acceptance at TMLR.